



The interference of tetrachloromethane in the measurement of benzene in air by Gas
Chromatography - Photoionization Detector (GC-PID).

**Authors:** Cristina Romero[1], María Esther González*[1], Marta Doval[2] and Enrique González[1]
[1]Chemical Engineering Department, School of Chemistry, University of Murcia, 30100 Murcia, Spain
[2]Chemical and Environmental Engineering Department, Technical University of Cartagena, Paseo
Alfonso XIII, 52, 30203 Cartagena, Murcia, Spain
*Correspondence to: María Esther González (esthergd@um.es)
**ABSTRACT**
The European Union requires that the benzene in air is measured due to its toxic characteristics and
widespread presence in the population nuclei, motivated by vehicle emissions. The reference
measuring technique is by gas chromatography (GC). For practical and safety reasons it is
recommended that a photometric ionisation detector (PID) is used. The automatic chromatographs
used in the monitoring stations must verify the operating conditions in the standard EN 14662:2005
part 3, which describes the Type Approval Tests. One of the tests determines possible
interferences, including by tetrachloromethane (TCM). Part 3 of the cited standard was modified in
2015, eliminating TCM as a possible interferer. Given that some studies ensure the presence of said
product in air, there has been considerable interest in testing different mixtures of benzene and
TCM.
This study has shown that the simultaneous presence of benzene and TCM causes a significant
decrease in the first GC-PID readings. For TCM concentrations of 0.7 µg/m$^3$ (typical of urban areas)
and 4.5 µg/m$^3$ (detected in the vicinity of landfills), the relative errors in benzene concentration
were 34 and 70%, respectively, which are of unacceptable quality for the measurement of benzene.
Possible mechanisms have been proposed to qualitatively and quantitatively explain what happens
in the PID with benzene alone and in the presence of TCM.
Given the significant impact of the interferer, it is important to open a discussion forum to address
this issue.



## INTRODUCTION

Benzene is a volatile organic compound (VOC). This group contains a diverse set of compounds with 15 carbon atoms or less, vapour pressures greater than 0.01 KPa at 20 ºC and boiling temperatures below 260 °C. Methane, organo-metallic compounds, carbon monoxide and carbon dioxide are excluded.

Benzene originates from natural emissions (from vegetation, oceans, soils, sediments, microbial decompositions and volcanoes) and anthropogenic emissions from fossil fuels (mainly from vehicles and to a lesser extent from the combustion of wood that is widely used in central and northern European countries for domestic heating). It is also present in tobacco smoke and in a wide range of industrial and household products (solvents, adhesives, paints and cleaning products), and is also a raw material for the synthesis of other products, such as dyes, detergents, plastics and explosives (Guenther et al.1995). .

In Europe, the greatest anthropogenic emission of benzene is from vehicles (Sarigiannis et.al 2011), with a contribution estimated at 80–85%, as shown in table 1. Currently, its presence is regulated in petroleum automotives by Directive 2009/33/CE, and there must be less than 1% by volume emitted.

Table 1. Contribution of the main anthropogenic sources of benzene in Europe.

Due to the chemical stability of benzene compared with most VOCs, its permanence in the atmosphere is high (with a half-life of 12 days) compared to other hydrocarbons of similar molecular mass. Consequently, it can be transported over long distances and degraded by OH radicals in the troposphere, forming phenol and glyoxal, among other compounds.

VOCs can affect human health, and benzene is a recognised inductor of leukaemia and also affects the central nervous system, deteriorates the immune system and damages genetic material. Benzene is the only VOC regulated in Europe in terms of air quality; the annual average limit is 5 $\mu g/m^3$ (and its determination in air is mandatory, especially in urban centers.

## BACKGROUND AND OBJECTIVES

The standardised methods in Europe for the measurement of benzene concentrations in air have been established by Directive 2008/50/EC and are described in the standard EN 14662 published in 2005, which is composed of three parts. The first two parts are still valid, but part 3 was modified in 2015. Each part describes the use of GC with a capillary column, but they differ in methods of sample collection and analytical automation. Part 1 of the standard EN 14662 describes the sampling of air by aspiration, using active carbon as an adsorbent and carrying out thermal desorption before the analysis. Part 2 differs in the desorption process, in which carbon sulphide is used. Carbon sulphide was the most widely used solvent for VOCs captured on carbonaceous adsorbents in industrial hygiene and atmospheric pollution testing. Currently, carbon sulphide is not used.




Part 3 of the 2005 and 2015 versions of the standard describes a method based on ambient air
sampling and automatic analysis, which is commonly used in measuring stations for atmospheric air
pollution in Europe. Both versions describe Type Approval that consists of a series of tests that are
carried out in accredited laboratories. In Europe, it is mandatory that air pollutant measuring
equipment complies with the aforementioned requirements before commercialisation.

The Type Approval tests of the two versions (2005 and 2015) of the cited standards and other
complementary tests were carried out in our laboratory under controlled conditions. Two articles
have been previously published regarding the effect of absolute air pressure at the entrance of the
chromatographs (Romero et al.2016), humidity and ambient temperature on the benzene
measurements (Romero et al.2017).

This article describes the possible interference of organic compounds that coexist in ambient air
with benzene. These interferences may have various causes, and highlight the following:

For a substance to act as an interfering agent, it must have a retention time in the chromatographic
column within the interval for the identification of benzene, so that both species reach the detector
within this interval.

If the above is true, the interference causes an increase or decrease in the detector signal. When
the chromatograph has a PID, one of the following occurs:

1. If any organic compound other than benzene is ionised by the radiation of the detector
lamp, the electronic density, and consequently the electrical signal, increase, which leads to
an increase in the concentration of benzene. For this, the ionization potential of the
interferent must be lower than that associated with the radiation of the lamp.

2. When the interferer causes a decrease in the benzene signal, there can be several causes:

2.1. The radiation of the detector lamp is absorbed to a greater or lesser extent by the
interferer, and the remaining energy is insufficient to completely ionise the benzene. This
phenomenon is known as the "quenching effect" (Chou 1999).This has also been described
and evaluated in a previous article (Romero et al.2017).

2.2. The interferer absorbs (blocks) part of the formed ions that participate in the
quantification of benzene, leading to decrease in the detected concentration. This
mechanism is known as quenching effect via electron capture. As discussed below,
tetrachloromethane (TCM) acts in this way.

The 2005 version of part 3 of the standard EN 14662 included a list of paraffinic, cyclic and
halogenated organic compounds (trichlorethylene and TCM) that should be tested as possible
interferers. In the 2015 version, all hydrogenated compounds are maintained, isooctane (2,2,4-
trimethylpentane) and 1-butanol have been added and TCM has been removed. Table 2 shows all
the common and specific components of each version.




136 Table 2. Organic compounds used to assess interference in the measurement of benzene in air, in
137 accordance with standards EN 14662-3 (2005 and 2015).


139 When we carried out the interference tests, part 3 of the standard (2005) was still current and the
140 Type Approval Test was carried out as indicated, where the established benzene concentrations
141 were close to 0.5 $\mu g/m^3$ and 40 $\mu g/m^3$. For these concentrations, benzene measurements were
142 compared in the absence and presence of the interfering agent (table 2). We observed that the
143 presence of the interfering agent did not affect the smaller benzene concentrations (approximately
144 0.5 $\mu g/m^3$) but the highest concentrations decreased significantly, up to a third of its value.

145

146 Locoge et al (2010) used the same chromatograph, obtaining the chromatograms as shown in figure
147 1, and observed the following:

148

149  1. Peak 2, which was within the interval that quantified benzene, also contained the following
150   compounds: cyclohexane, 2-methylhexane, 2,3-dimethyl pentane and TCM.
151  2. The first three cited interferers increased the concentration of benzene because their
152   respective ionisation potentials (between 9.88 and 10.08 eV) were lower than the potential
153   generated by the detector lamp (10.6 eV), and therefore the electric intensity in the
154   detector increased, leading to an increase in the apparent concentration of benzene.
155  3. TCM exhibited different behaviour as its ionisation potential (11.7 eV) was greater than
156   that emitted by the lamp; therefore it does not ionise or increase the intensity in the
157   detector. However, it is evident that it acts in an inverse manner, since it significantly
158   decreased the apparent benzene concentration.


161 Figure 1. Benzene chromatogram with the organic compounds described in the standard as
162 potential measurement interferers.


164 The aim of this article was to study the cited behaviour of TCM. First, the possible presence of this
165 compound in air that can lead to high deviations in benzene concentrations was studied. Second, a
166 mechanism was proposed to explain the aforementioned deviations.


168 The synthesis of TCM for direct use is prohibited by the Montreal Protocol because it is a substance
169 that destroys the ozone layer. However, its use as a raw material for the synthesis of other
170 substances such as hydrofluorocarbons (HFC), pyrenthroid pesticides or perchloroethylene is
171 allowed (Graziosi et al, 2016). Diffuse emissions may occur in its manufacture or during its use in
172 the aforementioned syntheses. In this sense, Penny et al (2010) estimated in 2007 emissions of,
173 approximately, 9500 MT of TCM in 192 countries. However, since the entry into force of the
174 Montreal Protocol, there has been a progressive decrease in the environmental presence of TCM,
175 with a decrease in the global average concentration of 10 to 15 pptv / decade (equivalent to 69 to
176 104 $ng/Nm^3$/decade). In 2005, Agency for Toxic Substances and Disease Registry (2005) determines
177 a global average concentration of 0.7 $\mu g/Nm^3$ with peaks in urban areas of 1.4 to 4.5 $\mu g/Nm^3$ and
178 Brosas (2008) detected 45 $\mu g/Nm^3$ in the vicinity of landfills. Most recent data provided by Blas et
179 al. (2016) in diverse cities of the world confirm average values of 0.61 $\mu g/Nm^3$ in Lukang (Taiwan),

0.64 µg/Nm$^3$ in Bristol (UK) and 1.10 µg/Nm$^3$ in Bilbao (Spain). In this last city, maximum
concentrations of 9.94 µg/Nm$^3$ have been measured. These values justify the concentrations that
have been used in the tests carried out in this work. It should also be noted that the TCM has a half-
life in the troposphere from 26 to 35 years (SPARC, 2013; Liang et.al, 2014), so its effects, such as
the one indicated here, will be manifested during the next two decades.
**EXPERIMENTAL**
**Test atmosphere:** A standard atmosphere system was used, which allowed "zero" air, and mixtures
of air with benzene and/or TCM with known concentrations, at a temperature of 20±2 $^o$C, pressure
of 1013 hPa and 50% relative humidity.
The airflows containing the components of interest (benzene and TCM) were prepared as in the
installation shown in figure 2, which consisted of the following:
• **Initial air:** Ambient air was obtained from a compressor free of lubrication oil to avoid air
pollution with organic volatiles contained in the oil (the compression chamber was made of
polytetrafluoroethylene (Teflon®), which did not require lubricant oil). The compressor dried the air
to 5% relative humidity, providing an adjustable pressure controller at the outlet that was
maintained at 1.8 bars.
• **"Zero" air purification and humidification:** After leaving the compressor, the air passed through
several beds containing silica gel, active carbon and alumina that almost completely dried it and
eliminated possible traces of organic and other inorganic contaminants (mainly nitrogen dioxide
and other acidic gases). After purification, the concentration levels of the organic pollutants in
"zero" air were periodically checked by chromatographic analysis, ensuring they were below the
limits of chromatographic detection. The airflow, up to a maximum of 200 L/min at atmospheric
pressure, was regulated with a Bronkhorst HI-T mass controller (B HI-T), depending on the required
concentrations of the components for each test.
Figure 2. Diagram of the standard atmosphere installation used to obtain airflows with benzene
and/or TCM at known concentrations
After purification, the airflow was humidified to a relative humidity of 50±5%, which was constant
for all tests, given its effect on the results. To humidify the airflow, the air conduction behind the
compressor was bifurcated: one through which dry air circulated and another that was connected
to a humidifier, until reaching airflow saturation. The regulation of both flow rates (dry and wet)
was ensured using two B HI-T mass controllers. Both lines were rejoined to obtain the final required
moisture level, measured with a hygrometer. The flow rate of water vapour supplied to the air after
the humidifier was calculated.
• **Incorporation of pollutants into the "zero" air stream:** The pollutants (benzene and TCM) mixed
with nitrogen were stored in bottles at certified concentrations, which were connected to the


humidified "zero" air line. The concentrations of benzene in the two bottles were 335 $\mu g/m^3$ and
1054 $\mu g/m^3$. The concentrations of TCM in the two bottles were 17.6 $\mu g/m^3$ and 65.3 $\mu g/m^3$.
The final standard concentration of pollutant x ($Cp_x$, expressed in $\mu g/m^3$) after mixing the flows of
the bottle/s and the air was determined from the mass balance as:
$Cp_x = Q_{bx} \cdot Cpb_x/(Q_a + \Sigma Q_b)$          Eq. (1)
where $Cpb_x$ is the concentration of x in the bottle of origin, $Qb_x$ is the flow of the bottle containing
pollutant x, and $Q_a$ is the humidified "zero" air flow and $\Sigma Q_b$ is the sum of the flows, respectively.
The above concentrations and flow rates were under the reference conditions of temperature and
pressure, 293 K and 101.3 KPa, respectively. These conditions were maintained at the entrance of
each chromatograph for all tests. There was a barometer in the laboratory, and high sensitivity
manometers were connected to the input of each chromatograph to maintain the flow at the
reference pressure.
In order to maintain the air currents at the reference temperature (20±2 $^o$C), the above-described
installation and the chromatographs were placed in respective chambers equipped with thermal
conditioning.
Two Syntech Spectras GC955 (Groningen, Netherlands) chromatographs were used. These are
widely used in European air pollution monitoring networks, and were identified as analysers I and
II.
The analytical process was semi-continuous. While the analyser was analysing a sample, a new one
was sampled and sent to the pre-concentration system.
The procedure is described in the following stages:
• The air sampling system comprised a 35 mL capacity piston pump, with 175 mL aspiration, and
the suction operation was repeated five times. The successive 35 mL samples of air moved to a pre-
concentrator (consisting of a column filled with Tenax), which retained the organic compounds and
released the excess air.
• Once the five suction cycles were completed, the contaminants retained in the pre-concentrator
underwent thermal desorption and were pulled with Nitrogen 5.0 (as a carrier gas) towards the
chromatographic column. The column was 15 m in length and 0.32 mm in diameter, and was
formed of silica with a film of adsorbent polymer (1 $\mu$m heliflex coating) for substances with boiling
points between 40 and 250 $^o$C. The initial oven temperature was set at 50 $^o$C and maintained for 3
min, then increased to 70 $^o$C at 10 $^o$C/min. This temperature was maintained for 7 min before being
reduced to 50 $^o$C with a cooling rate of 10 $^o$C/min.
• To explain the behaviour of benzene in the PID, we proposed the model shown in figure 3, which
also served as a basis to determine what happens in the presence of TCM (figure 5). When the



gasified benzene (n, in molar units) leaves the column, pulled by the carrier gas, it accesses the PID
where a fraction, F (≤1), is ionised by the radiation of the lamp, forming nF ionic couples (electrons
and benzyl cations). This forms a magma that produces an electrical intensity when passing through
the electrodes of the detector, whose area is proportional to the concentration of benzene in the
sample, given that F is practically constant within the range of concentrations tested, as
demonstrated by the experimental results. The benzyl ions recover the electrons in the cathode
and benzene is reformed, hence the non-destructive nature of the detector.
• The "windows" used to quantify the benzene in chromatographs I and II used, were 176–212 s
and 148–182 s, respectively.
Figure 3. Model of the behaviour of benzene in the PID of the chromatograph in the absence of
TCM.
Each analytical determination (from the first aspiration of air to the final result of the detected
concentration) occurred in 15 min.
The measurements were obtained using the following sequence:
• A continuous flow of "zero air" was prepared with the relative humidity, temperature and
standard pressure mentioned above. The chromatographic analysis of at least three samples of air
was performed until free of organic components.
• Once the absence of contaminants was verified, the flow of benzene was added to the "zero" air
from their respective bottles and, where appropriate, the TCM. Flows were previously calculated to
obtain the final concentrations required in the tests. In each experiment, at least six samples were
taken after measuring the stabilities of the concentration levels, and the average was determined.
**RESULTS AND DISCUSSION**
According to the standard EN 14662, the parameters used to evaluate the deviations caused by the
interferers were:
• Effect of organic compounds, $b_{Corg}$:
$$b_{Corg} = \left| C_{aCorg} - C_a \right| / C_a \quad \text{Eq. (2)}$$
where $C_{aCorg}$ is the average concentration of benzene chromatographic measurements in the
presence of organic compounds ($\mu g/m^3$) and $C_a$ is the average concentration of individual benzene
measurements in the absence of organic compounds ($\mu g/m^3$).
• The typical uncertainty, $U_{Corg}$:




$$U_{Corg} = \mid C_{aCorg} - C_a \mid / \sqrt{3} \quad \text{Eq. (3)}$$

• The test value, $V_{test}$:
$$V_{test.} = (U_{Corg} / C_a) \cdot 100 \quad \text{Eq. (4)}$$


We have included the relative error (RE):

$$RE = b_{Corg} \cdot 100 \quad \text{Eq. (5)}$$

**Previous experiments**

First, the organic interference tests were carried out. These tests have been established in the
earlier version of part 3 of the standard EN 14662, and involve comparing the responses of the
chromatographs when analysing standards ($C_p$) containing benzene at two concentrations (one-
tenth of the limit value established in the European legislation, Directive 2000/69/CE, and close to
70–90% of the maximum certification range) and mixtures of benzene and the organic compounds
indicated in table 3, each with concentrations close to 10 $\mu g/m^3$. The results are shown in table 3
using the previously defined parameters.

In each test, six individual measurements were made to obtain statistically significant data, using
the arithmetic mean as a representative value ($C_b$).

Table 3. Results obtained in the tests for the interference of organic compounds in benzene for
analysers I and II, with respect to those established in part 3 of the standard EN 14662.

As shown in table 3, the mixture of organic compounds interfered significantly, causing errors close
to 60% for the highest concentrations of benzene in the two chromatographs tested.

Based on the results, an analogous test was performed for concentrations of benzene close to the
limit value established in the Community Legislation (5 $\mu g/m^3$). The results are shown in table 4.

Table 4. Results for the test of the interference of organic compounds in benzene for analysers I
and II, for benzene concentrations close to the limit value in Europe.

The results also showed the interference of the mixture of organic compounds on the readings of
benzene concentration, though in this case the deviations were different for each chromatograph.

Due to the different behaviours of organic compounds that reach the PID in the benzene "window"
(see Background and Objectives), separate studies should be carried out for those that positively
(increasing) and negatively (decreasing) affect the concentration of benzene. TCM caused the
concentration of benzene to decrease, and was interferer studied in this article.





**Effect of TCM on benzene measurement**

To study the effect of TCM on the chromatographic measurements of benzene, the analyzer was first calibrated ($C_a$ versus $C_p$) to standard concentrations ($C_p$) of the pollutant, approximately 0, 2.5, 5, 10, 20 and 40 $\mu g/m^3$.

Next, air flows containing benzene were prepared at the same concentrations but with the addition of TCM flows diluted with nitrogen, such that the final concentrations of TCM ($C_{TCM}$) were 0.5, 1.0, 2.0 and 5.0 $\mu g/m^3$, leading to the chromatographic reading ($C_{aTCM}$). The TCM concentrations were selected due to the presence in urban areas at these levels (section 2), allowing estimations of the possible deviations in expected benzene measurements. To establish the concentrations of the benzene and TCM standards in the test chamber shown in table 5, the proportions of the different streams (zero air, and the benzene and TCM bottles diluted with nitrogen) have been taken into account, and in this case, equation 1 was applied. In all cases, each measurement was repeated six times and the standard deviations of each measurement are indicated in parentheses.

Table 5. Concentrations of benzene without TCM ($C_a$) and with TCM ($C_{aTCM}$), measured by analyzer I for different concentrations of TCM

From the results in table 5, the presence of TCM significantly decreased the benzene concentration readings with respect to the standards, and the deviations increased with TCM concentration. Likewise, for the same TCM concentration, the relative errors of the readings at different standard benzene concentrations remained practically constant.

The experimental values of $C_a$ and $C_{aTCM}$ versus $C_p$ of table 5 are shown in figure 4, and the following was observed:

1. The relationship between $C_a$ or $C_{aTCM}$ and $C_p$ for each series was linear (p<0.001) and passed through the origin of the coordinates, leading to the following general equations:

$$C_a = K^* C_p \text{ (without TCM) Eq. (6)}$$
$$C_{aTCM} = K C_p \text{ (with TCM) Eq. (7)}$$

where $K^*$ and $K$ are the slopes of the respective straight lines.

2. The slopes of the straight lines decreased with increasing TCM concentration.

Figure 4. Representation of benzene concentrations, $C_a$ and $C_{aTCM}$, read by analyser I, compared with the introduced patterns, $C_p$, at different TCM concentrations.

Table 6 shows the adjusted equations of the experimental values in table 5. A decrease in K was observed as the concentration of the interferent increased.

Table 6. Adjusted equations for the obtained experimental values.





To attempt to explain what occurs in the PID, and find the best relationship in terms of K for TCM
concentrations, we proposed the model in figure 5. The sequence and behaviour of the two
analysed species are depicted as they pass through the detector. The basic concepts of the model
are as follows:

1. The air sample, containing benzene (n moles) and TCM (m moles), accesses the PID. The lamp
ionises a fraction F of benzene but does not act on the TCM, since its ionisation potential is greater
than that provided by the lamp.

2. When the magma is formed, the mechanisms that take place are complex, given that the
electrons formed by benzene ionisation (nF) are distributed between two competing paths. One
part (pF) is directed towards the anode of the detector, and the other (qF) is retained by the strong
electronegativity of TCM. Thus, the measurement by the detector (pF) depends on the electric
fields configured by both systems and the quantities of benzene (n) and TCM (m). This may cause
one of the species to be limiting, this is that one of them is in default with respect to the other,
which also affects the distribution.

3. Finally, the system changes as shown in figure 5. The electric circuit closes and the initial species
is regenerated, showing that the PID is non-destructive in nature.

Figure 5. Simplified model of the behaviour of benzene and TCM when interacting simultaneously
in the PID detector

From a quantitative point of view, the proposed model and experimental data and establish the
following:

• According to figure 3, the concentration of benzene read by the chromatograph in the absence of
TCM ($C_a$) is expressed by:

$$C_a = C_p = n.F.M_b /V_T = n.F.\alpha_b \qquad \text{Eq. (8)}$$

where $M_b$ is the molecular mass of benzene, $V_T$ is the volume of the air sample and $\alpha_b = M_b/V_T$.

• As explained previously, when benzene and TCM simultaneously coexist, the benzene
concentrations read by the chromatograph ($C_{aTCM}$) for a given concentration of TCM, followed the
generic representation in figure 6, the following equation was confirmed:

$$C_{aTCM} = K.C_p = C_p - \Delta \qquad \text{Eq. (9)}$$

where $\Delta$ is the deviation of $C_{aTCM}$ from $C_p$.

Figure 6. Generic representations of benzene concentrations ($C_a$ and $C_{aTCM}$) read by the
chromatograph with respect to the standard concentrations ($C_p$)





As can be deduced from figures 4 and 6, $\Delta$ was proportional to $C_p$ for each $C_{TCM}$, and was also
dependent on $C_{TCM}$, which was determined experimentally. Based on the above, the following
function was proposed:

$\Delta = C_p . \varphi(C_{TCM})$        Eq. (10)


• Based on figures 5 and 6 and equations 9 and 10:

$C_{aTCM} = p.F.\alpha_b = (n - q).F.\alpha_b = C_p - q.F.\alpha_b = C_p - \Delta = C_p - C_p.\varphi(C_{TCM}) = [1 - \varphi(C_{TCM})].C_p$    Eq. (11)


From equations 9 and 11:

$1 - K = \varphi(C_{CTM})$              Eq. (12)


Table 6 shows the values of $1 - K$, which correlated with the respective TCM concentrations. The
best fit is represented by the following equation:

$1 - K = 0.389.C_{TCM}^{0.388}$    ($r^2 = 0.988$)   Eq. (13)


From equations 9, 11 and 13:

$C_{aTCM} = (1 - 0.389.C_{TCM}^{0.388}) C_p$    Eq. (14)

From equation 14, the relative error of the benzene chromatographic measurements in the
presence of TCM (RE) can be estimated:

$RE = [(C_{aTCM} - C_p)/C_p].100 = (0.389.C_{TCM}^{0.388}).100$         Eq. (15)


Thus, for TCM concentrations of 0.7, 1.4 and 4.5 µg /m$^3$ (levels that are currently found in urban
areas), errors may occur in benzene readings close to 34, 44 and 70%, respectively. These
deviations are high and not acceptable.
As indicated in the standard EN 14662:2005, TCM was included as a possible interfering
contaminant to be evaluated, but was not included in the new version. However, section 8 of the
current standard establishes that *"some compounds, including carbon tetrachloride or butanol, may*
*be present under site-specific conditions. In such cases, the responsibility for the proper*
*determination of benzene falls on the network that operates the analyzer by the appropriate choice*
*of separation conditions (analytical column, temperature program of the column)"*. This approach
seems difficult for network managers to implement, and we feel that the manufacturers of the
equipment should be responsible for solving this problem since they have the required technology
and equipment.
Therefore, a forum should be opened to discuss this problem, given that the GC-PID equipment is
the most widely used in the EU for the measurement of benzene in air and it would be advisable to
determine the best way to resolve this issue.


**CONCLUSIONS**

1. Given the toxic characteristics of benzene, the EU has established that its determination in ambient air is mandatory, particularly in urban areas. Benzene forms part of automotive fuels and is also present during the combustion of these fuels, leading to the establishment of an annual average limit value of 5 $\mu g/m^3$. GC-PID has also been confirmed as the reference technique for the detection of benzene, as described in the standard EN 14662:2005. This standard comprises three parts. Part 3 was modified in 2015, and describes the tests (so-called Type Approval Tests) that must be conducted and passed by the automatic measurement equipment before being marketed in EU countries. One such test verifies whether organic compounds that are common in ambient air interfere with the measurement of benzene, including TCM. This compound was considered in the earlier 2005 version of the standard but was excluded from the new 2015 version.

2. The research described in this article determined that the measurement of benzene by GC-PID in the presence of TCM caused a significant decrease in the concentration of benzene. The relative error (RE) of the concentration of benzene measured as a function of the concentration of TCM ($C_{TCM}$) can be calculated from the following expression:

$$RE = (0.389.C_{TCM}^{0.388}).100$$

Thus, for $C_{TCM}$ values of 0.7 $\mu g/m^3$ (typical of urban areas) and 4.5 $\mu g/m^3$ (in the vicinity of landfills), the REs in benzene concentration would be 34 and 70%, respectively, which are independent of the concentration of benzene.

3. Given the importance of this behaviour, a possible mechanism was proposed to explain the phenomenon when benzene is measured in the presence and absence of the interferent.

4. Of note, it is established in part 3 of the standard EN 14662:2015 that the managers of the air pollution monitoring network are responsible for determining the presence of TCM in the area where the atmospheric pollutants are measured. If detected, they must act to eliminate the effect of the interferent. We believe that the manufacturers of the chromatographs have greater technical and scientific capacity to solve this issue than the network managers.

This study highlights the uncertainty of measuring benzene using a GC-PID, and it is important to open a forum for discussion of this issue.

REREFENCES.

ATSDR. Public Health summary of CCl$_4$. Atlanta: US Public Health Services, 2005.

Blas, M., Uria-Tellaetxe, I., Gomez, M.C., Navazo, M., Alonso, L., García, J.A., Durana, N., Iza, J., Ramon, J.D. Atmospheric carbon tetrachloride in rural background and industry surrounded urban areas in northern Iberian Peninsula: Mixing ratios, trends, and potential sources 562, 26-34. https://dx.doi.org/10.1016/j_scitotenv2016.03.177, 2016.



Brosas, D. Ambient air monitoring of hazardous volatile organic compounds in Seattle, Washington
neighborhoods: trends and implications. Doctoral Thesis. Department of Civil Engineering and
Environment.Washington University. USA, 2008.
Carleton, J., Evenson, KM. Rate constants for the reactions of OH with ethane and some halogen
substituted ethanes at 296 K. Journal of Chemical Physics 64, 4303-4306.
https://doi.org/10.1063/1.432115, 1976.
Chou,J. 'Hazardous gas monitors: A practical guide to selection, operation and applications, 1st
edn'. McGraw-Hill Professional, 1999.

Clyne, MAA., Holt, PM. Reaction-Kinetics involving ground X2-PI and exited A2-XI+ Hydroxyl
radicals. 2. Rate constants for reactions of OH X2-PI with halogenomethanes and halogenoethanes.
Journal of the chemical society-Faraday transactions II 75,582-591, 1979.
Cox, RA.,Derwent, RG., Holt, PM. Relative rate constants for reactions of OH radicals with H-2, CH4,
CO, NO and HONO at atmospheric-pressure and 296 K. Journal of the chemical society-Farady
Transactions I 72, 2031-2043. doi:10.1039/F19767202031, 1976.

Directive 2008/50/EC of the European Parliament and of the Council of 21 May 2008 on ambient air
quality and cleaner air for Europe, Off. J. Eur. Communities, 152, 1–43. http://eur-
lex.europa.eu/LexUriServ/LexUriServ.do? uri=OJ:L:2008:152:0001:0044:EN:PDF (last access: 10 april
545 2018).

Directive 2009/33/EC of the European Parliament and of the Council of 23 April 2009 on the
promotion of clean and energy-efficient road transport vehicles.
ELI: http://data.europa.eu/eli/dir/2009/33/oj (last access: 10 april 2018).
Graziosi, F., Arduini,J., Bonasoni, P., Furlani, F., Giostra, U., Manning, A.J., McCulloch, A., O'Doherty,
s., Simmonds, P.G., Reimann, S., Vollmer, M.K., Maione, M. Emissions of carbon tetrachloride from
Europe.Atmospheric Chemistry and Physics 16, 12849-12859.doi: 10.5194/acp-16-12849-2016.

Guenther,A., Hewitt, CN., Erickson, D., Fall, R., Geron, C., Graedel, T., Harley, PLK., Lerdau, M.,
McKay. WATP., Sholes, B., Steinbrecher, R., Tallmraju, R., Taylor, J., Zimmerman, P. A global model
of natural volatile organic compounds emissions. Journal of Research 100, 8873-8892.
doi:10.1029/94JD02950, 1995.

Liang, Q., Newman, P.A., Daniel, J.S., Reimann, S., Hall, B.D., Dutto, G., and Kuijpers, L.J,M.
Constraining the carbon tetrachloride ($CCL_4$) budget using its global trend and inter-hemispheric
gradient. Geophys.Res.Lett., 41, 5307-5315. Doi:10.1002/2014GL060754, 2014.

Lillian,D., Singh, HB., Appleby, A., Lobban, L., Arnts, R., Gumpert, R., Hague, R., Toomey, J., Kazazis,
J., Antell ,M. Fates and levels of ambient halocarbons. ACS Symposium Series 17,152-158, 1975.



Locoge,N., Léonardis, T., Mathé, F. Analytical characteristics for benzene and VOC automatic
measuring system: results from laboratory tests and field campaign. 1stWorshop–AirMonTech
London, 2010.
Penny, C., Vuilleumier, S., Bringel, F. Microbial degradation of tetrachloromethane:mechanisms and
perspectives for bioremediation. FEMS Microbiology Ecology 74, 257-275. Doi:10.1111/j.1574-
6941.2010.00935.x, 2010.
Romero,C., Doval, M., González Duperón, E., González, E. Study of the effect of sample pressure on
*in situ* BTEX chromatographs. Environmental Monitoring and Assessment 188, 1-8.
doi:10.1007/s10661-016-5674-8, 2016.
Romero, C., Doval, M., González Duperón, E., González, E. Influence of simple temperature and
environmental humidity on measurements of benzene in ambient air by transportable GC-PID.
Atmospheric Measurement Techniques 10 , 4013-4022. https://doi.org/105194/amt-10-4013-2017,
580 2017.

Sarigiannis , DA., Karakitsios, SP., Gotti, A., Liakos, IL., Katsoyiannis. A Exposure to major
volatile organic compounds and carbonyls in European indoor environments and associated health
risk. Environment International 37,743-765. doi: 10.1016/j.envint.2011.01.005, 2011.
Singh, HB., Viezee, W., Johnson, WB., Ludwig, FL. The impact of stratospheric ozone on
tropospheric air-quality. Journal of the air pollution control association 30, 1009-1017.
https://doi.org/10.1080/00022470.1980.10465139, 1980.
SPARC: SPAC Report on the Lifetimes of Stratospheric Ozone Depleting Substances, Their
Replacements, and related Species, edited by: K0, M., Newman, P., Reimann, S., and Strahan, S.,
SPARC Report No. 6, WCRP-15/2013, available at: http://www.sparc-
climate.org/publications/sparc-reports/sparc-report-no6/ , 2013.
Standard EN 14662-1. Ambient air quality. Standard method for measurement of benzene
concentrations. Part 1: Pumped sampling followed by thermal desorption and gas
chromatography, 2005.
Standard EN 14662-2. Ambient air quality. Standard method for measurement of benzene
concentrations. Part 2: Pumped sampling followed by solvent desorption and gas chromatography,
596 2005.

Standard EN 14662-3. Ambient air quality. Standard method for measurement of benzene
concentrations. Part 3: Automated pumped sampling with *in situ* gas chromatography, 2005.
Standard EN 14662-3. Ambient air - Standard method for the measurement of benzene
concentrations - Part 3: Automated pumped sampling with in situ gas chromatography, 2015.



Table 1. Contribution of the main anthropogenic sources of benzene in Europe.

| Source | Percentage (%) |
|---|---|
| Vehicles | 80-85 |
| Oil refineries | 0.3-1.5 |
| Fuel distribution | 2.6-6 |
| Chemical industry | 1.3-13 |
| Domestic heating | 3-7 |
| Use of solvents | 1-4 |


Table 2. Mixture of organic compounds to assess interferences in the measurement of benzene in
air, in accordance with standards EN 14662-3 (2005 and 2015).

| EN 14662-3:2005 | EN 14662-3:2015 |
|---|---|
| Methylcyclopentane | |
| 2,2,3- Trimethylbutane | |
| 2,4-Dimethylpentane | |
| Cyclohexane | |
| 2,3-Dimethylpentane | |
| 2-Methylhexane | |
| 3-Ethylpentane | |
| Trichlorethylene | |
| n-Heptane | |
| Tetrachloromethane | 1-Butanol |
| | 2,2,4-Trimethylpentane |


Table 3. Results obtained in the test of the interference of the organic compounds in the benzene
readings for the analyzers I and II, according to what is established in part 3 of the standard EN
614 14662.

| ANALYZER I | | | | | |
|---|---|---|---|---|---|
| Standard concentrations introduced | | $C_b$ (µg/m³)[1] | RE (%) | $U_{Corg}$ (µg/m³) | $V_{test}$ (%) |
| $C_p$ C₆H₆ (µg/m³) | $C_p$ organic compounds (µg/m³) | | | | |
| 0.50 | 0.00 | $C_a$: 0.48 (0.04) | | | |
| 0.00 | 10.00 | 0.05 (0.00) | 2.25 | 0.01 | 1.30 |
| 0.50 | 10.00 | $C_{aCorg}$: 0.49 (0.04) | | | |
| 32.55 | 0.00 | $C_a$:33.07 (0.25) | | | |
| 0.00 | 10.00 | 0.05 (0.00) | 60.71 | 11.59 | 35.05 |
| 32.55 | 10.00 | $C_{aCorg}$: 13.00 (1.05) | | | |
| ANALYZER II | | | | | |
| Standard concentrations introduced | | $C_b$ (µg/m³)[1] | RE (%) | $U_{Corg}$ (µg/m³) | $V_{test}$ (%) |
| $C_p$ C₆H₆ (µg/m³) | $C_p$ organic compounds (µg/m³) | | | | |
| 0.50 | 0.00 | $C_a$: 0.48 (0.04) | | | |
| 0.00 | 10.00 | 0.05 (0.00) | 3.37 | 0.01 | 1.95 |
| 0.50 | 10.00 | $C_{aCorg}$: 0.50 (0.03) | | | |
| 39.50 | 0.00 | $C_a$:39.58 (0.25) | | | |
| 0.00 | 10.00 | 0.05 (0.00) | 60.56 | 13.84 | 34.97 |
| 39.50 | 10.00 | $C_{aCorg}$: 15.61 (0.36) | | | |

[1]The parenthesis shows the standard deviation of the 6 measurements.





Table 4. Results obtained in the test of the interference of organic compounds in the benzene
readings of analyzers I and II for benzene concentrations close to the limit value in Europe.

| ANALYZER I | | | |
|---|---|---|---|
| Standard concentrations introduced | | $C_b$ $(\mu g/m^3)^1$ | RE (%) |
| $C_p$ $C_6H_6$ $(\mu g/m^3)$ | $C_p$ organic compounds $(\mu g/m^3)$ | | |
| 4.68 | 0.00 | $C_a$: 4.64 (0.02) | |
| 0.00 | 10.00 | 0.05 (0.00) | 69.2 |
| 4.68 | 10.00 | $C_{aCorg}$:1.43 (0.08) | |
| ANALYZER II | | | |
| Standard concentrations introduced | | $C_b$ $(\mu g/m^3)^1$ | RE (%) |
| $C_p$ $C_6H_6$ $(\mu g/m^3)$ | $C_p$ organic compounds $(\mu g/m^3)$ | | |
| 5.06 | 0.00 | $C_a$: 5.03 (0.17) | |
| 0.00 | 10.00 | 0.05 (0.00) | 40.4 |
| 5.06 | 10.00 | $C_{aCorg}$:3.00 (0.01) | |

[1]The parenthesis shows the standard deviation of the measurements.







Table 5. Concentrations of benzene without TCM ($C_a$) and with TCM ($C_{aTCM}$), read by Analyzer I for
different concentrations of TCM.

| Series I: $C_{TCM}$ = 0.5 µg / m³. | | | |
|---|---|---|---|
| $C_p$ $C_6H_6$ (µg/m³) | $C_a$ (without TCM) (µg/m³) | $C_{aTCM}$ (with 0.5 µg/m³ de TCM) (µg/m³) | RE (%) |
| 0.00 | 0.00 (0.00) | -0.01 (0.00) | - |
| 1.15 | 1.17 (0.01) | 0.90 (0.05) | 22.90 |
| 3.48 | 3.45 (0.03) | 2.43 (0.01) | 29.56 |
| 8.62 | 8.55 (0.15) | 6.10 (0.14) | 28.61 |
| 22.25 | 20.19 (0.12) | 14.32 (0.12) | 29.07 |
| 42.60 | 42.57 (0.28) | 31.32 (0.16) | 26.42 |
| Series II: $C_{TCM}$ = 1.0 µg/m³ | | | |
| Cp $C_6H_6$ (µg/m³) | $C_a$ (without TCM) (µg/m³) | $C_{aTCM}$ (with 1 µg/m³ de TCM) (µg/m³) | RE(%) |
| 0.00 | 0.00 (0.00) | -0.01 (0.00) | - |
| 1.25 | 1.21(0.01) | 0.72 (0.00) | 40.33 |
| 3.55 | 3.45 (0.02) | 2.03 (0.03) | 41.16 |
| 8.70 | 8.49 (0.09) | 5.01 (0.06) | 40.99 |
| 20.31 | 20.22 (0.13) | 11.88 (0.05) | 41.25 |
| 42.89 | 43.01 (0.19) | 25.68 (0.07) | 40.29 |
| Series III: $C_{TCM}$ = 2.0 µg/m³ | | | |
| Cp $C_6H_6$ (µg/m³) | $C_a$ (without TCM) (µg/m³) | $C_{aTCM}$ (with 2 µg/m³ de TCM) (µg/m³) | RE(%) |
| 0.00 | 0.00 (0.00) | -0.01 (0.00) | - |
| 2.49 | 2.26 (0.01) | 1.00 (0.01) | 55.75 |
| 5.00 | 5.07 (0.02) | 2.18 (0.03) | 57.00 |
| 11.32 | 11.40 (0.11) | 4.64 (0.04) | 59.30 |
| 23.77 | 23.85 (0.11) | 10.19 (0.24) | 57.27 |
| 42.49 | 42.57 (0.28) | 20.95 (0.10) | 50.79 |
| Series IV: $C_{TCM}$ = 5.0 µg/m³ | | | |
| Cp $C_6H_6$ (µg/m³) | $C_a$ (without TCM) (µg/m³) | $C_{aTCM}$ (with 5 µg/m³ de TCM) (µg/m³) | RE(%) |
| 0.00 | 0.00 (0.00) | -0.01 (0.00) | - |
| 3.35 | 3.41 (0.2) | 1.18 (0.01) | 65.40 |
| 5.56 | 5.73 (0.03) | 1.97 (0.02) | 65.62 |
| 10.01 | 9.86 (0.10) | 2.88 (0.05) | 70.79 |
| 20.04 | 19.80 (0.14) | 5.88 (0.10) | 70.30 |
| 40.02 | 40.42 (0.18) | 11.87 (0.09) | 70.63 |











Table 6. Adjusted equations of the experimental values obtained.

| Series | $C_{TCM}$ ($\mu g/m^3$) | Calibration $C_a = K^* C_p$ ($r^2$) | $C_{aTCM} = K C_p$ ($r^2$) | 1-K |
|---|---|---|---|---|
| I | 0.00 | $C_a = 0.980 C_p$ (0.997) | --- | --- |
|  | 0.50 | --- | $C_{aTCM} = 0.715 C_p$ (0.995) | 0.285 |
| II | 0.00 | $C_a = 1.00 C_p$ (0.999) | --- | --- |
|  | 1.00 | --- | $C_{aTCM} = 0.595 C_p$ (1.00) | 0.405 |
| III | 0.00 | $C_a = 1.00 C_p$ (1.00) | --- | --- |
|  | 2.00 | --- | $C_{aTCM} = 0.474 C_p$ (0.992) | 0.526 |
| IV | 0.00 | $C_a = 1.01 C_p$ (1.00) | --- | --- |
|  | 5.00 | --- | $C_{aTCM} = 0.297 C_p$ (0.998) | 0.703 |




















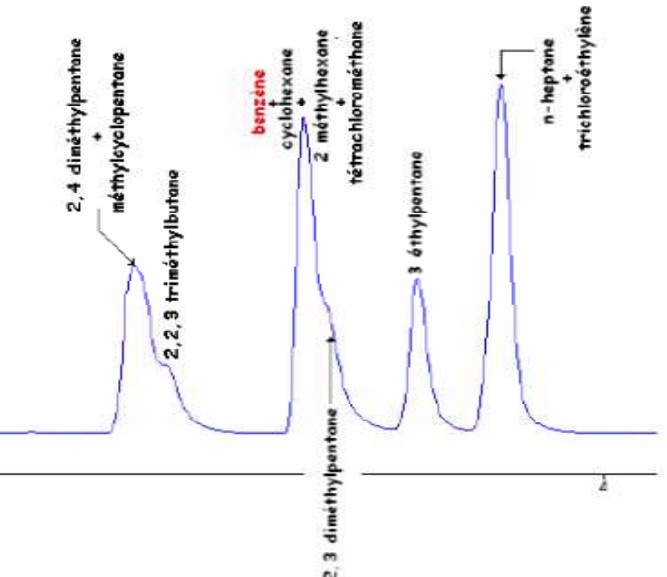

Figure 1. Benzene chromatogram with the organic compounds considered in the standard as
possible interferences in the measurement. (Locoge et al.,2010)

Figure 2. Diagram of the components of the standard atmosphere installation used to obtain air
flows with benzene and/or TCM at known concentrations.





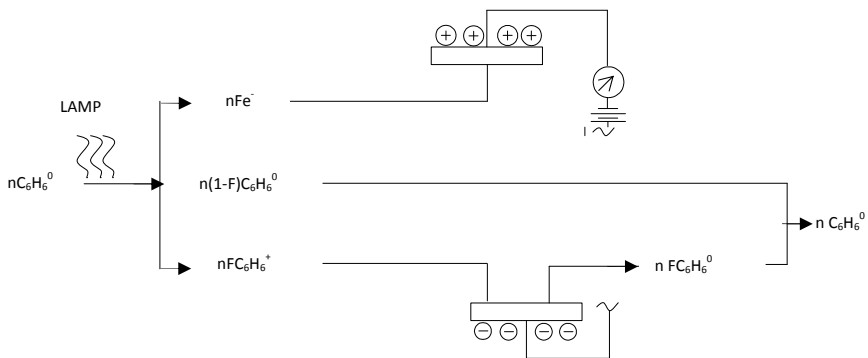

Figure 3.  Behaviour model of benzene in the PID of the chromatograph in absence of TCM.

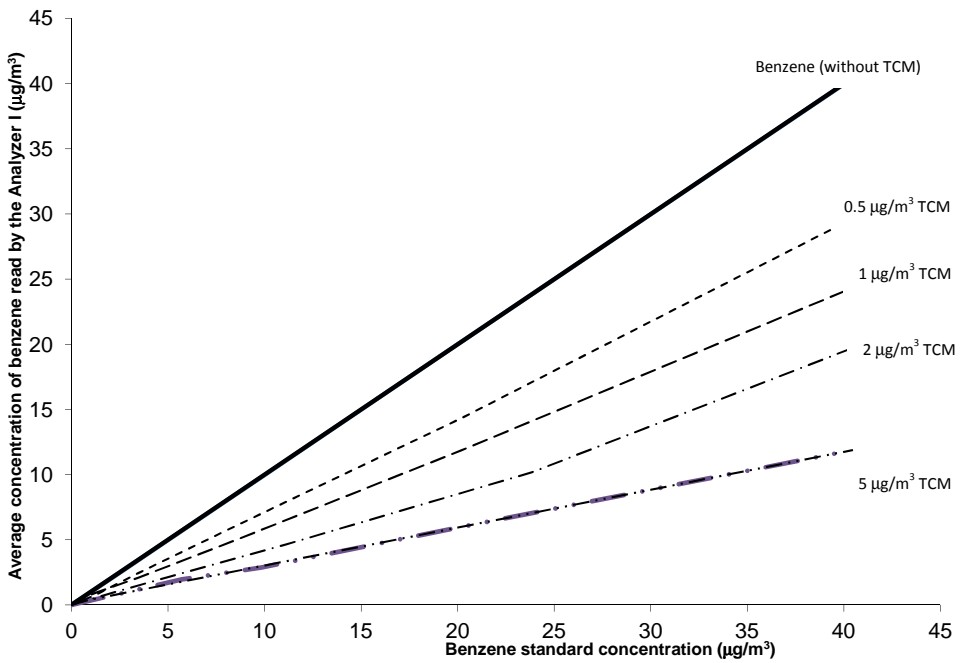

Figure 4. Analyzer calibration lines for benzene without and with various concentrations of TCM.

Figure 5. Simplified model of the behavior of benzene and TCM when they interact simultaneously in the PID detector.





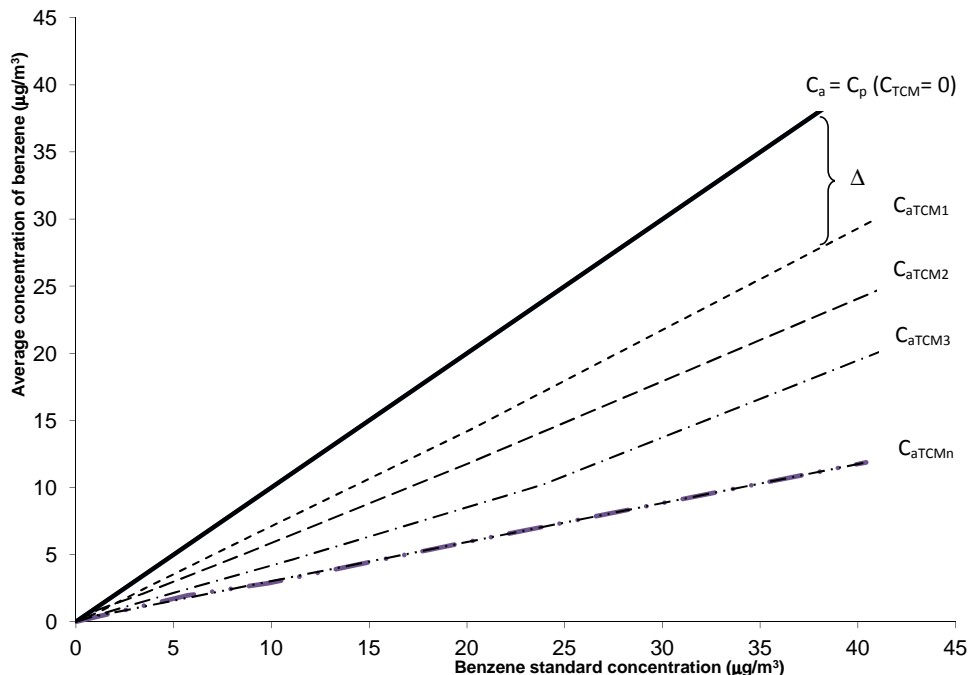

2  Figure 6. Generic representations of benzene concentrations read by chromatograph ($C_a$ and
3  $C_{aTCM}$) with respect to the standard concentrations introduced ($C_p$).
4
