# Peer review of "The interference of tetrachloromethane in the measurement of benzene in air by Gas Chromatography - Photoionisation Detector (GC-PID)"

_Atmospheric Measurement Techniques, 2018_

## Referee Comment (RC1) · Anonymous Referee #1 · 14 Dec 2018

Line 18.- Is there is an official recommendation? – Please provide a reference. Line 33.- Please be more pragmatic in the resolution of this problem. Apart from requesting the manufacture intervention, it would be useful to present a list of measurements to carry out by the user in order to minimize or avoid this problem. Line 49.- change degrees to °C Line 59.- Remove double endpoint. Line 70.- Why is it compared with hydrocarbons of similar molecular weight? This is not an indicator of stability in the atmosphere. Line 76 .- Please make reference to the corresponding legislation. Line 127.- Please provide appropriated reference. Line 223.- Who was the certifying body?

What are the uncertainties of the final generated concentrations? Line 356.- This seems a relevant item to be reported in the conclusions to be considered in the EN standard. Line 474.- Although these biases seem very high, it would be of interest to demonstrate that they are significant compared to the measurement uncertainties by considering the whole experimental setup. Line 486 .- Please consider my comments on Line 33. Instead of proposing a discussion forum, for which the revision of this paper in the public domain provide you with such a possibility, it is expected from the authors some directions on the issue. (i.e. new test proposal to the EN standard or other solving approaches). Line 614.- Why are Ucorg and Vtest(%) reported only in this Table? Line 649.- Why are the results of analyzer II not reported? Line 470 and 649.- What is the reproducibility of the Eq(15) between different analyzers?
* * *

---

## Referee Comment (RC2) · Anonymous Referee #3 · 11 Jan 2019

This manuscript discusses possible interferences in the measurements of benzene using a European Standard method. The manuscript fits in AMT topic wise. There are no errors in the manuscript but it is a report rather than a scientific manuscript. The manuscript needs to be substantially restructured and reformatted to even fit the instructions for AMT (See https://www.atmospheric-measurement-techniques.net/for_authors/manuscript_preparation.html ) especially in regards to sections. The manuscript preparation also lacks attention to detail and is sloppy. The present webpublished version has randomly underlined sections, which might be track

changes from previous editing. This is very sloppy and together with numerous typos, one can only hope the authors performed their scientific work more carefully than the preparation of their manuscript.

Some detailed comments (typos/language issues are incomplete and formatting needs to be appropriate for a journal article)

L18 - A PID in English analytical literature is a Photoionization detector not a photometric ionization detector.

L33-34 appropriate for a conclusion/outlook but not for an abstract.

L49 why is the degree underlined? Why is there random underlining of text and references?

L59 excess period

L61 and check rest of the manuscript it is "et al." not "et. al" overall format references AND in text citations following author instructions.

L63 provide references/citations to the directive document Table1: provide in the table (or table legend) where these values are form (reference?)  as this is not from the present work.

L69 Statements like lifetime need to be supported by citations. Overall the whole introduction needs much more citations for the statements made (like L73 VOCs can affect human health.. based on what?).

L76 missing parenthesis.

L78 Why is there a section on introduction and background and objectives.  There should only be an introduction.  Overall the manuscript is not prepared using AMT instructions

L81-83 and following: provide citations for standards

L150 please write chemical names correctly.. dimethylpentane (one word)

L153 what is the electric density? Do you mean electron density?

L185 it is typical in a scientific manuscripts to end the intro with 2-3 sentences of what is in the manuscript.

Experimental: info on source/ manufacturer purity of gases, purification material etc is critically missing.. please provide basic info on used reagents and used instrumentation including manufacturer location etc (see standard for the journal), Examples: L225 how were the concentrations obtained? Where are these gases from? Or L245 this is the only place where you give location etc as manufacturer info. This needs to be the case for everything, including the actual analyzers used!

L254 why are bullet points used here? This is not common and not really warranted. Again this is a scientific manuscript not a report.

Figure 5 seems to be used before figure 4. Make sure to use figures in the right order and number accordingly.

L325 what is the meaning of previous experiments. You show some of them here.. so this is not really previous? May be you men "initial". Also if this is published already, you need full citation.

L413 what is a magma? Did you mean "plasma" which would be the word for a partially ionized gas? Please use correct scientific language or explain what you mean? It is very uncommon to talk about a plasma (or magma) in the context of a PID.

The following discussion is very weak as it is qualitative and not quantitative. What is a "strong electronegativity"? This has no real meaning.

L484-486 this not results and discussion but a statement that ends a conclusion

Conclusions: This is not how conclusions are presented in a manuscript. For starters it is unusual to have numbered conclusions. Please write this like a conclusion in a

scientific manuscript. In addition your first conclusion point 1. is not in this manuscript. This is not a conclusion of this manuscript

L517: Proofread your manuscript! Use a spell checker!!!

Figure 1 quality is not acceptable: resolution, is too low

Tables should be uniformly formatted

English should be edited for better reading flow

―――――――――――――――――――

---

## Author Response (AR1)

**Reviewer 1:**

First of all, we would like to acknowledge the comments provided by Reviewer#1, which have help us improve our manuscript. Clarifications of the issues below have been included in the MS.

Line 18.- Is there is an official recommendation? – Please provide a reference

Thanks for the advice. The official recommendation is EN 14662 Standard. Ambient air - Standard method for the measurement of benzene concentrations - Part 3: Automated pumped sampling with in situ gas chromatography, 2015. However, we have decided to eliminate the complete sentence from the Abstract as it is not relevant.

Line 33.- Please be more pragmatic in the resolution of this problem. Apart from requesting the manufacture intervention, it would be useful to present a list of measurements to carry out by the user in order to minimize or avoid this problem.

We appreciate the suggestion from the reviewer. This matter has been clarified in the MS: we have added information in the Result and Discussion section about measurements to carry out by the user in order to minimize this problem. The final wording is "[…] *This approach would require continuous measurements of TCM in air and a knowledge of how TCM deviates measurements from its real value, which in turn, requires carrying out tests similar to those presented in this paper with dynamic dilution systems in controlled test atmospheres. This measure could not be easy to apply for economic and technical reasons so the whole responsibility must not only fall on the network managers. It seems reasonable that the manufacturers of the equipment tackle actions for solving this problem –or, at least, for reducing the extent of the interference in their measurements-, since they have the required technology and equipment. In any case, users of this type of equipment should be aware of the problem to try to minimise it. The discussion of this issue in the appropriate forum (e.g. the European Committee for Standardisation) seems also pivotal to reduce the uncertainty in benzene measurements by GC-PID in presence of TCM concentrations*".

Line 49/50.- change degrees to ºC:

It has been done. Thank you for your suggestion.

Line 59.- Remove double endpoint

Double endpoint has been deleted. Thank you for your recommendation.

Line 70.- Why is it compared with hydrocarbons of similar molecular weight? This is not an indicator of stability in the atmosphere

Thank you for this interesting comment. We agree with the reviewer and accordingly, the mention to hydrocarbons of similar molecular weight has been removed. It was a comment recommended by a previous journal reviewer.

Line 76 .- Please make reference to the corresponding legislation

We have added the reference: Directive 2008/50/EC of the European Parliament and of the Council of 21 May 2008 on ambient air quality and cleaner air for Europe, Off. J. Eur. Communities, 152, 1–43.

http://eurlex.europa.eu/LexUriServ/LexUriServ.do?uri=OJ:L:2008:152:0001:0044:EN:PDF

Line 127.- Please provide appropriated reference.

Thank you for your suggestion. The following reference has been added: Senum, G. I.: Quenching or enhancement of the response of the photoionization detector, J. Chromatogr. A, 205(2), 413–418, doi:10.1016/S0021-9673(00)82668-0, 1981.

Line 223.- Who was the certifying body?

We appreciate the suggestion from the reviewer; we have included in the MS the certifying bodies, which were the respective manufacturers of the gas mixtures. The mixtures were certified according to Standard ISO 6141:2000.

What are the uncertainties of the final generated concentrations?

This is an interesting point. Flow rates were continuously controlled and the expanded uncertainty of the generated concentrations was calculated from the standard uncertainty of the gas mixtures in the gas cylinders and the standard uncertainties of the flow rates. We have added a new paragraph in the Experimental set-up section: "*The expanded uncertainties of all generated concentrations of pollutants were estimated from the standard uncertainties of the high concentration gas cylinders and the standard uncertainties of the gas flow rates. In all cases, the final expanded*

*uncertainty was less than 5%, according to the limit established in Standard EN 14662-3"*.

Line 356.- This seems a relevant item to be reported in the conclusions to be considered in the EN standard.

We agree with the reviewer. We have modified the Conclusion section accordingly: "The research described in this article has determined that TCM causes a significant interference in the measurement of benzene by GC-PID. This interference is negative, that is, readings of benzene are below their real ambient values, which may originates a mismanagement of the air quality of a location with presence of TCM in its air in relation to benzene".

Line 474.- Although these biases seem very high, it would be of interest to demonstrate that they are significant compared to the measurement uncertainties by considering the whole experimental setup.

As stated before, the uncertainty of the generated gas mixtures was below 5%. In addition, a lack-of-fit test was performed in order to test the accuracy of the readings. For this, after calibration of the analysers, several gas mixtures of benzene in air with different concentrations ranging from 0 to 50 µg/m$^3$ were measured. Relative differences between the readings and the reference concentrations were calculated and, in all cases, were below 10%. This value is much lower than the biases that occur in the readings when there is TCM in air ambient (34, 44 and 70 % when there is a TCM concentration of 0.7, 1.4 and 4.5 µg/m$^3$). Moreover, other potential influencing parameters such as temperature or pressure were kept constant during the experiments, allowing to conclude that the high biases obtained in readings when TCM was added to the mixture are due to the presence of this substance. The following paragraph has been included in the MS: "*In order to ensure that the biases obtained in these and subsequent experiments were only due to the interfering compounds tested, sample and surrounding temperatures, sample pressure and voltage were kept constant during all experiments. A lack-of-fit test was performed in order to test the accuracy of the readings. For this, after calibration of the analysers, several gas mixtures of benzene in air with different concentrations ranging from 0 to 50 µg/m$^3$ were measured. Relative differences between the readings and the reference concentrations were calculated and, in all cases, were below 10%, much lower than the values reported in the Result and Discussion section*".

We have added a new paragraph in the "Results and discussion section" (please see comment on line 33): Also, in "Conclusions" we have added a similar conclusion. The final wording is the following: "*Interestingly enough, it is established in part 3 of the standard EN 14662:2015 that the managers of the air quality monitoring network are responsible for determining the presence of TCM in the area where benzene is measured. If detected, they must act to eliminate the effect of the interferent. However, this approach would require continuous measurements of TCM in air and a knowledge of how TCM deviates measurements from its real value, which in turn, requires carrying out tests similar to those presented in this paper with dynamic dilution systems in controlled test atmospheres. This may entail economic and technical issues so manufacturers of the chromatographs should try to solve this problem as they have greater technical and scientific capacity than network managers. In any case, all these issues should be discussed in the appropriate forum (e.g. the European Committee for Standardisation) in order to improve the uncertainty of benzene measurements and, thus, the management of the air quality*".

Parameters $U_{Corg}$ and $V_{test}$ (%) were only reported in Table 3 because this table contained the data obtained according to concrete indications of the EN 14662 Standard, which requires the calculation of $V_{test}$ and $U_{Corg}$ in order to compare and verify its acceptability with the performance criterion (<5%) established in such standard.

We consider that using the relative error is more logical and useful to evaluate the deviations than the $V_{test}$ and $U_{Corg}$ parameters. For this reason, in the rest of the Tables, which contained results of tests proposed by ourselves and not contemplated in the Standard, such parameters were not included ($U_{Corg}$ and $V_{test}$). However, we have decided to merge former Tables 3 and 4 into a single new table (Table 2) in order to save space, and we have decided to calculate parameters $V_{test}$ and $U_{Corg}$ for the results presented in former Table 4 to maintain the symmetry of new Table 2.

Line 649.- Why are the results of analyzer II not reported?

The Analyzer II belongs to the official surveillance network from the Government of Región de Murcia and it is operating continuously in a monitoring station. Therefore, we only had such equipment for a limited time in our lab. Given that both analysers exhibited a similar performance in the first set of experiments carried out, we

considered that the results obtained with Analyser I would be representative of both of them. In addition, a reproducibility test was carried out in the lab. Both analysers were subject to measure simultaneously a gas mixture containing 5 μg/m$^3$ nominal benzene in zero air. The reproducibility (in μg/m$^3$) was calculated as:

$$Reproducibility = \sqrt{\frac{\sum d_i^2}{2n}}$$

where $d_i$ is the $i$ difference in readings between Analyser I and II and $n$ is the total number of measurements (6 in our case). The value obtained was 0.067 μg/m$^3$ when the average concentration of benzene in the reference gas mixture gas 4.6 μg/m$^3$, which means 1.5% bias. This value was considered low enough to perform the subsequent tests just with one analyser. All these results have been added to the MS.

Line 470 and 649.- What is the reproducibility of the Eq (15) between different analyzers?

Eq. (15) was only obtained for Analyser I, as only this analyser was subject to the tests in Section 2.2.2. Reproducibility has been calculated as detailed in the previous comment.

**Comments to Reviewer#3**

This manuscript discusses possible interferences in the measurements of benzene using a European Standard method. The manuscript fits in AMT topic wise. There are no errors in the manuscript but it is a report rather than a scientific manuscript. The manuscript needs to be substantially restructured and reformatted to even fit the instructions for AMT (See https://www.atmospheric-measurement-techniques.net/for_authors/manuscript_preparation.html ) especially in regards to sections. The manuscript preparation also lacks attention to detail and is sloppy. The present webpublished version has randomly underlined sections, which might be track changes from previous editing. This is very sloppy and together with numerous typos, one can only hope the authors performed their scientific work more carefully than the preparation of their manuscript.

First of all, we the authors would like to apologise for the original manuscript. It is true that was sloppy and there was much room for improvement. We did misregard its format (though not its content) because we had an internal deadline to meet. In any case, we reiterate our apologies and we would really appreciate that Reviewer#3 agrees to review the new version of the manuscript, whose quality is much higher. Thank you very much in advance.

Some detailed comments (typos/language issues are incomplete and formatting needs to be appropriate for a journal article)

L18 - A PID in English analytical literature is a Photoionization detector not a photo-metric ionization detector.

Thank you for spotting this error. This has been corrected in the manuscript.

L33-34 appropriate for a conclusion/outlook but not for an abstract.

This two lines have been removed. The new abstract is as follows: "*The European Union requires that benzene in air is continuously measured due to its toxicity and widespread presence in the population nuclei, mainly motivated by vehicle emissions. The reference measuring technique is gas chromatography (GC). Automatic chromatographs used in monitoring stations must verify the operating conditions established in Standard EN 14662 part 3, which includes a type approval section with a number of tests that analysers must pass. Among these tests, the potential interference of a number of compounds is evaluated. The 2005 version of the mentioned standard requires the evaluation of the potential interference of tetrachloromethane (TCM). The 2015 version eliminates TCM as a potential interferent. Although most consumer uses of TCM have been banned, recent studies have measured significant concentrations of TCM in air. In this paper, the potential interference of TCM on benzene measurements obtained with gas chromatography coupled to a photoionisation detector (PID) has been investigated. Our study shows that the simultaneous presence of benzene and TCM causes a significant decrease in benzene readings. For TCM concentrations of 0.7 µg m$^{-3}$ (typical of urban areas) and 4.5 µg m$^{-3}$ (detected in the vicinity of landfills), the relative errors in benzene measurements were 34 and 70 %, respectively, which are far too high compared to the maximum overall uncertainty allowed for benzene measurements (25 %). Possible mechanisms to qualitatively and quantitatively explain the behavior of the PID when measuring benzene with and without TCM have been proposed*".*

L49 why is the degree underlined? Why is there random underlining of text and references?

Before it was sent to the reviewers, the editor handling the manuscript asked for making some changes in the manuscript. They were underlined to ease their locations in the text. All these underlines have now been removed.

L59 excess period

The excess period has been removed.

L61 and check rest of the manuscript it is "et al." not "et. al" overall format references AND in text citations following author instructions.

References throughout the text have been managed with Mendeley and formatted to Copernicus citing style, so now there is consistency in all of them and follow the journal instructions.

L63 provide references/citations to the directive document

This reference has been added.

Table1: provide in the table (or table legend) where these values are form (reference?) as this is not from the present work.

Table 1 has been removed as it reported data that was not up to date and it was not relevant for the research.

L69 Statements like lifetime need to be supported by citations. Overall the whole introduction needs much more citations for the statements made (like L73 VOCs can affect human health.. based on what?).

Introduction section has been thoroughly revised. References have been added in all statements done. The two first paragraphs of the introduction are the following ones and can give an idea of the whole work done throughout the manuscript.

"*Benzene is a volatile organic compound (VOC) (Tisserand and Young, 2014). Directive 2008/50/EC (European Commission, 2008) defines them as organic compounds from anthropogenic and biogenic sources, other than methane, that are capable of producing photochemical oxidants by reactions with nitrogen oxides in the presence of sunlight. Benzene sources include natural emissions from vegetation and oceans (Misztal et al., 2015), microbial decomposition (Neves et al., 2005), wildfires (Wentworth et al., 2018) and volcanoes (Tassi et al., 2015); and anthropogenic emissions mainly from vehicles that use fossil fuels (von Schneidemesser et al., 2010) and, in central and northern European countries, from the combustion of wood used for domestic heating (Hellén et al., 2008). It is also present in tobacco smoke (Darrall et al., 1998) and in a wide range of industrial and household products (solvents, adhesives, paints and cleaning products), and is also a raw material for the synthesis of other products, such as dyes, detergents, plastics and explosives (Guenther et al., 1995). Its content in gasoline is regulated by Directive 2009/33/CE and it has to be < 1 % (v/v) (European Commission, 2009).*

*Due to the chemical stability of benzene compared with most VOCs (with a half-life of 9.4 days (Atkinson, 2000)), its permanence in the atmosphere is high. Consequently, it can be transported over long distances. It is degraded by OH radicals in the troposphere, forming phenol and glyoxal, among other compounds (Atkinson, 2000; Volkamer et al., 2001). Benzene is a recognised inductor of leukaemia (D'Andrea and Reddy, 2016) and also affects the central nervous and immune systems and damages genetic material (Bahadar et al., 2014). Benzene is the only VOC whose concentrations in air are regulated in Europe. Its annual limit value is 5 µg m$^{-3}$ at 293 K and 101.4 kPa and its monitorisation in air is mandatory (European Commission, 2008)".*

L76 missing parenthesis.

This has been corrected.

L78 Why is there a section on introduction and background and objectives. There should only be an introduction. Overall the manuscript is not prepared using AMT instructions

These two sections have been merged into a single one and some of its content has been moved to other sections. The organisation of the paper has significantly changed and has adopted a typical structure of a scientific paper. AMT instructions for authors have been followed and a proper template has been used for the preparation of the revised version of the manuscript.

L81-83 and following: provide citations for standards

Citations for standards have been added:

CEN: EN 14662-1 Ambient air quality. Standard method for measurement of benzene concentrations. Pumped sampling followed by thermal desorption and gas chromatography, 2005a.

CEN: EN 14662-2 Ambient air quality. Standard method for measurement of benzene concentrations. Pumped sampling followed by solvent desorption and gas chromatography, 2005b.

CEN: EN 14662-3 Ambient air. Standard method for the measurement of benzene concentrations. Automated pumped sampling with in situ gas chromatography, 2005c.

CEN: EN 14662-3 Ambient air. Standard method for the measurement of benzene concentrations. Automated pumped sampling with in situ gas chromatography, 2015.

CEN: EN 14662-4 Ambient air quality. Standard method for measurement of benzene concentrations. Diffusive sampling followed by thermal desorption and gas chromatography, 2005d.

CEN: EN 14662-5 Ambient air quality. Standard method for measurement of benzene concentrations. Diffusive sampling followed by solvent desorption and gas chromatography, 2005e.

L150 please write chemical names correctly.. dimethylpentane (one word)

These names have been corrected. Thank you for spotting these mistakes.

L153 what is the electric density? Do you mean electron density?

We apologise for the mistake. These have been changed to *electric current*, which is the proper magnitude measured in the detector.

L185 it is typical in a scientific manuscripts to end the intro with 2-3 sentences of what is in the manuscript.

The new introduction section ends with the following lines: "*Given the above, in this paper, the potential interference of TCM on benzene measurements carried out by GC-PID is studied. A mechanism that explains the observed behaviour is also proposed*".

Experimental: info on source/ manufacturer purity of gases, purification material etc is critically missing.. please provide basic info on used reagents and used instrumentation including manufacturer location etc (see standard for the journal), Examples: L225 how were the concentrations obtained? Where are these gases from? Or L245 this is the only place where you give location etc as manufacturer info. This needs to be the case for everything, including the actual analyzers used!

Thank you for your comment. The experimental section has been revised and the maufacturers of equipment and reagents have been added. Also, the wording of the section has been changed and, hopefully, improved. As an example, the following is the first paragraph of the Experimental section:

"*An in-house designed controlled atmosphere chamber was used to generate dynamic test mixtures of benzene in air with and without TCM (Fig. 1). This chamber was used in previous works (Romero-Trigueros et al., 2016, 2017) and only a brief description will be given here. Zero air was generated from ambient air with a JUN-AIR compressor (Michigan, USA) provided with a drier, which is capable of reducing the relative humidity of the air down to 5 %. This dry air flows through three consecutive scrubbers containing silica gel (Merck, Darmstadt, Germany) and active charcoal (Chiemivall, Barcelona, Spain) to remove any traces of remaining humidity and other gases present in the air. After purification, a periodic check of organic pollutants in the zero air was carried out by gas chromatography, ensuring they were below their limits of detection. Benzene was incorporated to the zero air from a high-concentration gas mixture of benzene in nitrogen. Two mixtures of benzene in air from Abelló Linde (Valencia, Spain) were used at nominal amount fractions of 350 and 1000 µg m$^{-3}$ (5 % expanded uncertainty). TCM was also incorporated from one of the two gas cylinders of TCM in nitrogen available in the laboratory (18 µg m$^{-3}$ and 65 µg m$^{-3}$, 5 % expanded uncertainty) (Praxair, Guildford, UK) depending on the final concentrations required for our tests. All mixtures were certified by their respective manufacturers according to Standard ISO 6141:2000 (ISO, 2000). The flow rate of zero air and the target species were controlled and measured with Bronkhorst HI-TEC (Ruurlo, The Netherlands) mass flow controllers (ranges of 0–0.4 L min$^{-1}$ for the benzene in nitrogen and TCM in nitrogen mixtures and 0–12 L min$^{-1}$ for the zero air). The chamber allows for humidification of the mixtures with an in-house designed humidifier (Romero-Trigueros et al., 2017). Sample and environmental temperature can also be controlled as well as sample pressure at the inlet of the GC-PID. The laboratory was provided with a mercury barometer (Thies CLIMA, Göttingen, Germany), and high sensitivity Magnehelic gauges (Dwyer, Michigan, USA) were connected to the input of each chromatograph to maintain the flow at the reference pressure. Sample relative humidity and temperature were measured with a Testo 645 thermo hygrometer (Barcelona, Spain). All the tests carried out in the present work were done with dry gases at 20±2 ºC and 1013 hPa*".

L254 why are bullet points used here? This is not common and not really warranted. Again this is a scientific manuscript not a report.

Bullet points have been removed from all over the paper.

Figure 5 seems to be used before figure 4. Make sure to use figures in the right order and number accordingly.

Figures have been renumbered according to their order of appearence in the text.

L325 what is the meaning of previous experiments. You show some of them here.. so this is not really previous? May be you men "initial". Also if this is published already, you need full citation.

The *Experimental section* has been modified as well as the *Result and discussion* section. Now the *Experimental section* contains section *2.1. Experimental set-up,* and section *2.2. Experimental methods*. Inside the latter, section 2.2.1 describes the *experimental procedure to evaluate the potential interference of organic compounds according to Standard EN 14662:2005-3* (what was previously called "initial tests"). Section 2.2.2 describes the *experimental procedure to study the interference of TCM on benzene measurement*.

The *Result and discussion* section is organised following the two sub-sections of the *Experimental methods* section. Thus, inside section 3 *Results and discussion*, there is now two sub-sections: section 3.1 *Potential interference of organic compounds according to Standard EN 14662:2005-3*; and section 3.2. *Effect of TCM on benzene measurement*.

We the authors hope everything is clearer now.

L413 what is a magma? Did you mean "plasma" which would be the word for a partially ionized gas? Please use correct scientific language or explain what you mean? It is very uncommon to talk about a plasma (or magma) in the context of a PID.

We have to say that this is a clear "translation" error. We indeed mean the mixture with partially ionised gases. We have changed the word "magma" to "mixture".

The following discussion is very weak as it is qualitative and not quantitative. What is a "strong electronegativity"? This has no real meaning.

We apologise for this. We shouldn't have used electronegativity for a molecule as it is a property of bonded atoms. The correct magnitude is the electron affinity. A value of this parameter is also provided together with the reference from where it was obtained. The sentence has been changed to "*One part (pF) is directed towards the anode of the detector, and the other (qF) is retained by the TCM, given its relatively high electron affinity (2.2 eV) (Chen and Chen, 2004)*".

L484-486 this not results and discussion but a statement that ends a conclusion

We have ended the Result and discussion section with the following paragraph (we think that a bit of a discussion and some guidelines to try to solve the issue are appropriate in this section):

"*As indicated in the standard EN 14662:2005-3, TCM was included as a possible interfering contaminant to be evaluated, but was not included in the 2015 version of the Standard. However, Sect. 8 of the current standard establishes that "some compounds, including carbon tetrachloride or butanol, may be present under site-specific conditions. In such cases, the responsibility for the proper determination of benzene falls on the network that operates the analyser by the appropriate choice of separation conditions (analytical column, temperature program of the column)". This approach would require continuous measurements of TCM in air and a knowledge of how TCM deviates measurements from its real value, which in turn, requires carrying out tests similar to those presented in this paper with dynamic dilution systems in controlled test atmospheres. This measure could not be easy to apply for economic and technical reasons so the whole responsibility must not only fall on the network managers. It seems reasonable that the manufacturers of the equipment tackle actions for solving this problem –or, at least, for reducing the extent of the interference in their measurements-, since they have the required technology and equipment. In any case, users of this type of equipment should be aware of the problem to try to minimise it. The discussion of this issue in the appropriate forum (e.g. the European Committee for Standardisation) seems also pivotal to reduce the uncertainty in benzene measurements by GC-PID in presence of TCM concentrations*".

Conclusions: This is not how conclusions are presented in a manuscript. For starters it is unusual to have numbered conclusions. Please write this like a conclusion in a scientific manuscript. In addition your first conclusion point 1. is not in this manuscript. This is not a conclusion of this manuscript

Conclusion section has been thoroughly modified. The new section is as follows:

"*The research described in this article has determined that TCM causes a significant interference in the measurement of benzene by GC-PID. This interference is negative, that is, readings of benzene are below their real ambient values, which may originates a mismanagement of the air quality of a location with presence of TCM in its air in relation to benzene.*

*The relative error (RE) of the concentration of benzene measured as a function of the concentration of TCM ($C_{TCM}$) can be calculated from the following expression for the analyser tested in this work:*

$$RE = (0.389 \cdot C_{TCM}^{0.388}) \cdot 100$$

*Thus, for $C_{TCM}$ values of 0.7 µg m$^{-3}$ (typical of urban areas) and 4.5 µg m$^{-3}$ (in the vicinity of landfills), the REs in benzene measurements would be 34 and 70 %, respectively, independently of the concentration of benzene. These values are much higher than the overall expanded uncertainty allowed for benzene measurements with GC-PIDs. Given the importance of this interference, a possible mechanism has been proposed to explain the phenomenon when benzene is measured in the presence and absence of TCM. According to the proposed model, TCM attracts part of the electrons produced in the ionisation of benzene; thus, the electric current measured in the detector is lower than it should be. This interference is different in nature from that produced by other interfering species and, consequently, should be assessed independently of them.*

*Interestingly enough, it is established in part 3 of the standard EN 14662:2015 that the managers of the air quality monitoring network are responsible for determining the presence of TCM in the area where benzene is measured. If detected, they must act to eliminate the effect of the interferent. However, this approach would require continuous*

*measurements of TCM in air and a knowledge of how TCM deviates measurements from its real value, which in turn, requires carrying out tests similar to those presented in this paper with dynamic dilution systems in controlled test atmospheres. This may entail economic and technical issues so manufacturers of the chromatographs should try to solve this problem as they have greater technical and scientific capacity than network managers. In any case, all these issues should be discussed in the appropriate forum (e.g. the European Committee for Standardisation) in order to improve the uncertainty of benzene measurements and, thus, the management of the air quality*".

L517: Proofread your manuscript! Use a spell checker!!! Figure 1 quality is not acceptable: resolution, is too low Tables should be uniformly formatted English should be edited for better reading flow

Again, we are very sorry for the mistakes in the first version of the manuscript. We have proofread the new manuscript several times and have implemented all the changes suggested by the reviewers. Figure 1 has been removed as we have not been able to find it with better resolution. The information provided in Figure 1 has been described in the main text. Tables have been formatted according to AMT instructions. We hope the English style has also improved.

**The interference of tetrachloromethane in the measurement of benzene in air by Gas Chromatography - Photoionisation Detector (GC-PID).**

**Authors:** Cristina Romero-Trigueros[1], María Esther González*[2], Marta Doval Miñarro[3] and Enrique González[2]

[revised manuscript text omitted]

1157 Thus, based on Fig. (3) and Fig. (5) and Eqs. (9) and (10), $C_{aTCM}$ can be expressed as follows.

1159 $$C_{aTCM} = p.F.\alpha_b = (n - q).F.\alpha_b = C_p - q.F.\alpha_b = C_p - \Delta = C_p - C_p.\varphi(C_{TCM}) = [1 - \varphi(C_{TCM})].C_p \qquad \text{Eq. (11)}$$

1161 From Eqs. (9) and (11):
1162 $$1 - K = \varphi(C_{CTM}) \qquad \text{Eq. (12)}$$

1164 Table 4 shows the values of $1 - K$ for each TCM concentration tested. The best fit is represented by
1165 the following equation:

1167 $$1 - K = 0.389.C_{TCM}^{0.388} \quad (r^2 = 0.988) \quad \text{Eq. (13)}$$

User1 Anon 21/1/19 17:14
Figure 5. Simplified model of the behaviour of behaviour of benzene and TCM when interacting simultaneously in the PID detector . ... [67]

User1 Anon 21/1/19 17:20

User1 Anon 21/1/19 17:23

User1 Anon 21/1/19 17:30

User1 Anon 21/1/19 17:31

User1 Anon 21/1/19 17:32

User1 Anon 21/1/19 17:37

User1 Anon 21/1/19 17:37

User1 Anon 21/1/19 18:15

User1 Anon 21/1/19 18:15

User1 Anon 21/1/19 17:38

User1 Anon 21/1/19 18:16

User1 Anon 21/1/19 18:23

User1 Anon 21/1/19 18:23

User1 Anon 21/1/19 18:23

From Eqs. (9), (11) and (13):

$$C_{aTCM} = (1 - 0.389 \cdot C_{TCM}^{0.388}) \, C_p \quad \text{Eq. (14)}$$

[revised manuscript text omitted]

User1 Anon 6/2/19 14:56
User1 Anon 25/1/19 14:46
User1 Anon 21/1/19 18:47
User1 Anon 25/1/19 10:06
User1 Anon 25/1/19 14:54
User1 Anon 25/1/19 14:54
User1 Anon 25/1/19 14:53
User1 Anon 25/1/19 14:54
User1 Anon 25/1/19 14:54
User1 Anon 25/1/19 14:54
User1 Anon 25/1/19 14:54
User1 Anon 25/1/19 14:54
User1 Anon 28/1/19 20:32
User1 Anon 6/2/19 12:19
Usuario 24/1/19 15:36
… [103]

| 42.60 | 42.57 (0.28) | 31.32 (0.16) | 26.42 |
|---|---|---|---|

**Series II: $C_{TCM}$ = 1.0 $\mu g\ m^{-3}$**

| $C_p\ C_6H_6$ ($\mu g\ m^{-3}$) | $C_a$ (without TCM) ($\mu g\ m^{-3}$) | $C_{aTCM}$ (with 0.5 $\mu g\ m^{-3}$ de TCM) ($\mu g\ m^{-3}$) | RE(%) |
|---|---|---|---|
| 0.00 | 0.00 (0.00) | -0.01 (0.00) | - |
| 1.25 | 1.21(0.01) | 0.72 (0.00) | 40.33 |
| 3.55 | 3.45 (0.02) | 2.03 (0.03) | 41.16 |
| 8.70 | 8.49 (0.09) | 5.01 (0.06) | 40.99 |
| 20.31 | 20.22 (0.13) | 11.88 (0.05) | 41.25 |
| 42.89 | 43.01 (0.19) | 25.68 (0.07) | 40.29 |

**Series III: $C_{TCM}$ = 2.0 $\mu g\ m^{-3}$**

| $C_p\ C_6H_6$ ($\mu g\ m^{-3}$) | $C_a$ (without TCM) ($\mu g\ m^{-3}$) | $C_{aTCM}$ (with 0.5 $\mu g\ m^{-3}$ de TCM) ($\mu g\ m^{-3}$) | RE(%) |
|---|---|---|---|
| 0.00 | 0.00 (0.00) | -0.01 (0.00) | - |
| 2.49 | 2.26 (0.01) | 1.00 (0.01) | 55.75 |
| 5.00 | 5.07 (0.02) | 2.18 (0.03) | 57.00 |
| 11.32 | 11.40 (0.11) | 4.64 (0.04) | 59.30 |
| 23.77 | 23.85 (0.11) | 10.19 (0.24) | 57.27 |
| 42.49 | 42.57 (0.28) | 20.95 (0.10) | 50.79 |

**Series IV: $C_{TCM}$ = 5.0 $\mu g\ m^{-3}$**

| $C_p\ C_6H_6$ ($\mu g\ m^{-3}$) | $C_a$ (without TCM) ($\mu g\ m^{-3}$) | $C_{aTCM}$ (with 0.5 $\mu g\ m^{-3}$ de TCM) ($\mu g\ m^{-3}$) | RE(%) |
|---|---|---|---|
| 0.00 | 0.00 (0.00) | -0.01 (0.00) | - |
| 3.35 | 3.41 (0.2) | 1.18 (0.01) | 65.40 |
| 5.56 | 5.73 (0.03) | 1.97 (0.02) | 65.62 |
| 10.01 | 9.86 (0.10) | 2.88 (0.05) | 70.79 |
| 20.04 | 19.80 (0.14) | 5.88 (0.10) | 70.30 |
| 40.02 | 40.42 (0.18) | 11.87 (0.09) | 70.63 |

Table 4. Calibration lines of Analyser I obtained by linear regression (without TCM ($C_a$) and with TCM ($C_{aTCM}$)). $R^2$ is shown in brackets. Parameter (1-K) is also shown.

| Series | $C_{TCM}$ ($\mu g.m^{-3}$) | Calibration $C_a = K^* C_p$ ($R^2$) | $C_{aTCM} = K\ C_p$ ($R^2$) | 1-K |
|---|---|---|---|---|
| I | 0.00 | $C_a$ = 0.980 $C_p$ (0.997) | --- | --- |
|  | 0.50 | --- | $C_{aTCM}$ = 0.715 $C_p$ (0.995) | 0.285 |
| II | 0.00 | $C_a$ = 1.00 $C_p$ (0.999) | --- | --- |
|  | 1.00 | --- | $C_{aTCM}$ = 0.595 $C_p$ (1.00) | 0.405 |
| III | 0.00 | $C_a$ = 1.00 $C_p$ (1.00) | --- | --- |
|  | 2.00 | --- | $C_{aTCM}$ = 0.474 $C_p$ (0.992) | 0.526 |

| | | | | | |
|---|---|---|---|---|---|
| IV | 0.00 | $C_a = 1.01\ C_p$ | (1.00) | --- | --- |
| | 5.00 | --- | $C_{aTCM} = 0.297\ C_p$ (0.998) | | 0.703 |

[Figure]

Usuario 24/1/19 15:54

Unknown

[Figure]

[Figure]

[Figure]

User1 Anon 6/2/19 15:11

Figure 1: Schematic of the components of the controlled atmosphere chamber used to obtain gas mixtures of benzene in air with and without potential interferent substances. (HI: Humidity indicator, TI: Temperature indicator, PI: Pressure indicator)

User1 Anon 6/2/19 15:11

User1 Anon 6/2/19 15:11

Usuario 5/2/19 17:35

[Figure]

[Figure]

User1 Anon 6/2/19 15:09

Figure 2: Calibration lines for Analyser I with and without TCM at different concentration levels.

User1 Anon 25/1/19 15:00

**Comentario [1]:** Esta figura y la siguente hay que cambiarlas porque la figura tiene que ser un todo, no vale ponerle etiquetas con las concentraciones de cada curva encima del gráfico (como está ahora). Además hay que cambiar la forma de poner las unidades de la concentración a μg m$^{-3}$

Si puedes, pídele a Esther los archivos originales. Tampoco entiendo por qué hay una línea morada (la de 5 ug/m3)

User1 Anon 6/2/19 15:12

[Figure]

Figure 3: A generic representation of benzene readings as a function of the concentration of benzene and TCM in the reference gas mixture.

[Figure]

1821

1822 Figure 4. Schematic of the behaviour of benzene in the PID of the chromatograph in absence of
1823 TCM.

[Figure]

Figure 5: Schematic of the behaviour of benzene and TCM when they interact simultaneously in the PID detector.

User1 Anon 6/2/19 15:14

————Salto de sección (Página siguiente)————

---

## Author Response (AR2)

Murcia, 15.02.2019

Associate Editor

Atmospheric Measurement Techniques

Subject: Revised manuscript to *Atmospheric Measurement Techniques*

Dear Dr. Herckes,

Thank you very much for your quick reply and your valuable comments.

We have changed "analyzer" to "analyser" and used British English throughout the manuscript.

We have also modified the references and hopefully they are all consistent with Copernicus guidelines.

Finally, we have added information about the silica gel, the active charcoal and the purity of the carrier gas. We have spotted that a gas cylinder was missing from M&M section and we have included it. Regarding the gas mixtures of benzene, TCM and organic interferents, their respective certificates only state the certified concentrations for the target species with the corresponding expanded uncertainty and the balance gas (nitrogen) but there is no more information about impurities on them (probably they are below the detection limit of the analytical instrument used in the certification), so we have kept the information regarding these mixtures as it was. We hope this is satisfactory enough.

We are looking forward to hearing from you.

Sincerely yours,

Esther

[revised manuscript text omitted]